# Role of Nitric Oxide in the Altered Calcium Homeostasis of Platelets from Rats with Biliary Cirrhosis

**DOI:** 10.3390/ijms241310948

**Published:** 2023-06-30

**Authors:** Masoud Akbari Aghdam, Paola Romecín, Joaquín García-Estañ, Noemí M. Atucha

**Affiliations:** Departamento de Fisiología, Facultad de Medicina, Instituto Murciano de Investigación Biosanitaria, Universidad de Murcia, 30120 Murcia, Spainpaodunromec@gmail.com (P.R.);

**Keywords:** calcium signaling, bile-duct ligation, capacitative Ca^2+^ entry, liver cirrhosis, fura-2, thapsigargin, thrombin

## Abstract

Introduction: Previously, we found that intracellular calcium (Ca^2+^) homeostasis is altered in platelets from an experimental model of liver cirrhosis, namely the bile-duct-ligated (BDL) rat. These alterations are compatible with the existence of a hypercoagulable state. Objective: In the present study, we analyzed the role of nitric oxide in the abnormal calcium signaling responses of an experimental cirrhosis model, the bile duct-ligated rat. Methods: Chronic treatment with L-NAME was used to inhibit NO production in a group of control and BDL animals, and the responses compared to those obtained in a control and BDL untreated group (n = 6 each). The experiments were conducted on isolated platelets loaded with fura-2, using fluorescence spectrometry. Results: Chronic treatment with L-NAME increased thrombin-induced Ca^2+^ release from internal stores in both control and BDL rats. However, the effect was significantly greater in the BDL rats (*p* < 0.05). Thrombin-induced calcium entry from the extracellular space was also elevated but at lower doses and, similarly in both control and BDL platelets, treated with the NO synthesis inhibitor. Capacitative calcium entry was also enhanced in the control platelets but not in platelets from BDL rats treated with L-NAME. Total calcium in intracellular stores was elevated in untreated platelets from BDL rats, and L-NAME pretreatment significantly (*p* < 0.05) elevated these values both in controls and in BDL but significantly more in the BDL rats (*p* < 0.05). Conclusions: Our results suggest that nitric oxide plays a role in the abnormal calcium signaling responses observed in platelets from BDL rats by interfering with the mechanism that releases calcium from the internal stores.

## 1. Introduction

Hemostasis, in patients with cirrhosis, is influenced by multiple, opposing and changing variables, which can favor both the appearance of bleeding episodes and hypercoagulation states [1,2]. The platelet abnormalities described in cirrhosis include not only thrombocytopenia but also aggregation defects, either by a hyper or hypo-response [3,4]. Therefore, in coagulation disorders, an increase in factors that promote bleeding with procoagulant alterations that induce the appearance of thrombosis may coexist. Among the latter, we can mention the decrease in the activity of some anticoagulation and stasis mechanisms, vascular as a consequence of the slowing of the flow circulatory system and disorders of fibrinolysis and platelet activity. The analysis of the bibliography allows the verification of the existence of at least two types of coagulopathies, depending on the origin of the hepatic cirrhosis. Thus, it has been observed that patients with liver disease of cholestatic origin have a lower number of bleeding complications when compared to patients in a similar clinical state but with cirrhosis of viral etiology or alcohol. Although the origin of this alteration is not clear, the existence of a hypercoagulable state in patients with primary biliary cirrhosis has been suggested due to more effective platelet function in these patients than in others. Platelet activation requires an increase in cytoplasmic Ca^2+^ levels, which is reported to be defective in both experimental models of liver cirrhosis and patients [5,6,7]. In previous studies, platelets obtained from a rat model of biliary cirrhosis showed an increased amount of stored Ca^2+^ and an increased activity of SERCA [6].

The mechanisms responsible for these alterations in Ca^2+^ homeostasis in cirrhosis are not entirely clear, but it is likely that mediators such as nitric oxide (NO) are involved. The inhibitory effect of NO on platelet function was one of the first reported after its discovery [8]. Moreover, much evidence shows that excess NO plays an important role as a mediator of the pathophysiology of liver cirrhosis [9,10,11,12]. However, whether NO affects the defective calcium handling of platelets in biliary cirrhosis is unknown.

Thus, the objective of the present study was to investigate the role of NO in the alterations in platelet function observed in a rat model of biliary cirrhosis by chronically administering an inhibitor of NO synthesis, L-NAME. Specifically, we analyzed how the inhibition of NO synthesis affects the calcium signaling responses to agonist (thrombin) stimulation, the capacitative calcium entry that occurs after depletion of internal stores and the total amount of calcium available in these internal stores.

## 2. Results

Animals with bile duct ligation exhibited typical signs of liver cirrhosis, including jaundice, as well as an enlarged liver and spleen. Ascites was not present in any animal included in this study. Baseline Ca^2+^ levels were similar in platelets of control rats (21.6 ± 3.9 nM) and BDL rats (31.4 ± 4.2 nM), and they remained similar in platelets of L-NAME-treated rats (control, 25.5 ± 0.8 nM and L-NAME, 30.7 ± 0.7 nM).

Regarding intracellular Ca^2+^ release in response to thrombin, chronic L-NAME treatment increased the response after stimulation with thrombin (0.1 and 0.3 U/mL) in the absence of extracellular Ca^2+^ in both control and BDL rats (Figure 1). The response to thrombin 0.1 was of greater magnitude in the platelets of BDL rats (*p* < 0.05, 2.4 times vs. 1.8 times in controls) whereas with thrombin 0.3 the elevation was similar in both groups (1.6 times in controls and 1.9 times in BDL). After thrombin stimulation in the presence of calcium (Figure 2), L-NAME elevated the response but only at the dose of 0.1 U/mL (*p* < 0.05, 6.9 times in controls and 5.7 in BDL). The dose of thrombin 0.3 also elevated calcium levels, but this change was of a lower magnitude (1.1 times in controls and 0.9 in BDL) compared to the lower dose.

The response to thapsigargin administration in the absence of calcium (Figure 3) showed that the response was not increased in animals chronically treated with L-NAME, and the response in control platelets was reduced. However, the addition of calcium to these thapsigargin-pretreated platelets induced a massive entry of calcium that was significantly elevated (*p* < 0.05) in the L-NAME group (Figure 4) but only in the control animals (189 times compared to 64 times in the BDL platelets).

Total calcium in intracellular stores, as analyzed by the simultaneous addition of ionomycin and thapsigargin, was elevated in untreated platelets from BDL rats (6752.8 ± 359.4 nM in controls and 8698.5 ± 807.15 in BDLs), and L-NAME pretreatment significantly elevated these values (*p* < 0.05) up to 11,673.7 ± 449.1 nM in controls and 15,265.7 ± 925.4 in BDL platelets, significantly more in the BDL rats (*p* < 0.05).

## 3. Discussion

In previous studies [6,13,14], we demonstrated, in a rat model of liver cirrhosis induced by bile duct ligation, several platelet alterations consistent with a hypercoagulable state. These alterations are related to a defective platelet Ca^2+^ handling and are present before the appearance of ascites. In the present study, we confirm these alterations and extend them to show that NO plays a role in the defective calcium signaling observed in platelets of cirrhotic animals.

Conflicting results have been found when studying platelet function in cirrhosis, with reports indicating a higher risk of both thrombosis and bleeding [15]. In the experimental model used in the present study, we have also reported that enhanced aggregation and calcium responses are observed in the phase prior to the development of ascites, thus promoting thrombosis. In the ascitic phase, these platelet changes would induce bleeding alterations [6].

The present results show that, in platelets of both control and cirrhotic rats, in the absence of extracellular calcium, NO synthesis inhibition increased the amount of calcium released from intracellular stores, which would enhance platelet aggregation. Thus, confirming reports indicating that NO is a potent inhibitor of platelet adhesion and aggregation in cirrhotic animals [16]. However, in the presence of extracellular calcium, the stimulation with the agonist thrombin in the L-NAME-treated platelets was able to elevate calcium levels but only at the lowest concentration of the agonist and at the same level in both groups of animals. This clearly indicates that the primary effect of NO on calcium signaling is related to the inhibition of the release of calcium from intracellular stores, while the effect on entry from the extracellular space is evident only at lower, more physiological doses. This is in keeping with previous observations that report that, in relation to platelets, and in physiological doses, NO serves to maximize blood perfusion, prevent platelet aggregation and thrombosis as well as neutralize toxic oxygen radicals in the liver [17]. However, in the setting of chronic liver inflammation when a large sustained amount of NO is present, things may be different. 

We also analyzed the so-called capacitative calcium entry, which is known to play a central role in calcium signaling [6]. Essentially, calcium entry in the cytoplasm is regulated by the filling of the calcium stores, so when calcium is depleted or discharged from the stores, calcium entry from the external environment is promoted. Thus, when thapsigargin was used to deplete the internal stores, the experiments demonstrated that there is a defect in the mechanisms responsible for capacitative calcium entry, confirming our previous results [6], since the addition of calcium after thapsigargin administration, which inhibits calcium uptake into the intracellular stores, resulted in an increase in calcium only in the control platelets but not in the cirrhotic ones. We proposed that this defect may be related to an enhanced activity of SERCA, induced by NO. Thus, in a situation of high NO production, such as in liver cirrhosis, this excess of NO would stimulate SERCA activity in platelets, thus making more calcium available for release when platelet activation takes place. This excess of calcium then would participate in the enhanced coagulable state shown in the preascitic phase of the disease.

The objective of the paper was not to identify the source of NO but to find its role in the platelet alterations in this bile duct-ligated rat model. It is well known that both isoforms of NOS are involved in liver cirrhosis [18,19,20,21]. Thus, it appears that eNOS is involved in the characteristic hyperdynamic circulation and vasodilation of liver cirrhosis, playing a compensatory role. However, iNOS-derived NO contributes to pathological processes. We have also published that iNOS, as well as eNOS, is upregulated in smooth muscle cells isolated from the abdominal aorta of BDL rats [20]. Regarding platelets, it is also known that both isoforms are present in platelets. In platelets obtained from liver cirrhotic patients, the levels of NO synthase from both isoforms were elevated [21]. Thus, it is highly likely that, in our model, both isoforms contribute to the effects of NO. However, other effects related to the inhibitor used may be also contributing, since L-NAME has been reported to decrease brain-derived neurotrophic factor (BDNF) levels, which by affecting microcirculation in rats and by changing the levels of vasopressin may have a confounding effect in these responses [22,23]. This is a limitation of the present study and more studies with more specific inhibitors should be performed to better dissect the roles of the different NO isoforms. Other problems are related to the fact that the experiments were done in isolated and washed platelets which, although they are very frequently used in the laboratory setting, may interfere with the cell physiology. Thus, it would be interesting to perform other studies for instance, in whole blood, to study platelet function aggregation by means of light optical aggregometry, platelet-related primary hemostasis function using the PFA-100 System or the Multiplate Analyzer System. Also, the use of intravital microscopy to study platelet aggregation in vivo may be another useful technique to apply in order to circumvent the problems related to in vitro techniques [24]. 

In summary, in the present and our previous studies [6,13,14], performed in a rat model of liver cirrhosis by bile duct ligation, several platelet alterations compatible with the existence of a hyperaggregatory state were demonstrated. In general, these alterations are related to a defective platelet calcium handling, specifically to enhanced intracellular calcium release that is evoked by agonists and to an increased amount of calcium stored in the intracellular organelles, secondary to an enhanced activity of both smooth endoplasmic reticulum calcium ATP-ase (SERCA) and plasma membrane calcium ATP-ase (PMCA) [6] and are present before the appearance of ascites. After the appearance of ascites, calcium signaling in platelets is reduced so that calcium is not released in sufficient levels to induce platelet aggregation, thus favoring bleeding.

Thus, the findings of this study have important implications for understanding the mechanisms that underlie platelet dysfunction in liver cirrhosis and its potential impact on thrombosis and bleeding. The results indicate that NO plays a crucial role in defective calcium signaling observed in platelets of cirrhotic animals, which could contribute to the hypercoagulable state seen before ascites appears. However, these results do not have a direct translation to the clinics since they are of experimental nature, and thus, because of this, these conclusions should be taken with caution. Nevertheless, more studies would be necessary in order to go deep into the mechanism behind the capacitative calcium entry, for instance by analyzing the STIM1-Orai1 protein complexes that mediate the altered calcium response that our study has identified. Also, future investigations could aim to further elucidate the precise mechanisms by which NO affects calcium handling in platelets, as well as to explore the contribution of both eNOS and iNOS isoforms to these effects. Additionally, future studies should consider examining other factors beyond L-NAME inhibition that may be contributing to these responses, such as BDNF levels or vasopressin changes. Another important area of research would be the study of the release of platelet vesicles and its components, specifically the role of its products released, which at the moment it is not known in liver cirrhosis. Other studies in whole blood would be also interesting, as stated earlier, to eliminate the possible confounding interference of the experimental protocol with isolated and washed platelets. Finally, given the conflicting reports on platelet function in liver cirrhosis patients regarding their risk for both thrombosis and bleeding complications, future research using animal models can help clarify underlying pathophysiological mechanisms leading to altered hemostasis seen clinically.

## 4. Materials and Methods

Drugs: All the products used were from Sigma, except where indicated. Fura-2 AM (fura 2 acetoxymethyl ester, Molecular Probes, Eugene, OR, USA), Thapsigargin (Invitrogen, Madrid, Spain) was dissolved in DMSO. Appropriate dilutions were prepared freshly every day in measurement buffer.

Animals: Male Sprague-Dawley rats were obtained from the Animal House of the Universidad de Murcia and used for this study. All experiments were conducted in accordance with the ethical guidelines for laboratory animal treatment established by the European Union and were approved by the Ethics Committee of the Universidad de Murcia.

Experimental groups: Male Sprague-Dawley rats weighing approximately 250 g were divided into two groups—those who underwent bile-duct ligation (BDL) and excision and those who underwent sham operation (controls), as previously described [11,12]. Six animals per group were used in the present study. They were given normal rat chow and tap water ad libitum. Experiments were conducted between 21 and 28 days later, on animals showing no ascites. A group of control and BDL rats were chronically treated with N(ω)-nitro-L-arginine methyl ester (L-NAME), 1 mg/kg/day in their drinking water, up to the day of the experiment. L-NAME is a non-specific nitric oxide synthase inhibitor, which reduces the production of NO from both isoforms. The experiments were conducted in accordance with the ethical rules for the treatment of laboratory animals of the European Union and were approved by the Ethics Committee of the Universidad de Murcia. 

Isolation of platelets, fura-2 loading and determination of [Ca^2+^]i: The isolation of platelets, fura-2 loading and determination of [Ca^2+^]i were carried out as previously described [13,14,25,26]. Briefly, animals were anesthetized with a mixture 1:1 of ketamine and xylazine hydrochloride, 0.2 mL/100 g body weight, intramuscular, and blood was obtained from the abdominal aorta in a plastic tube containing an anticoagulant solution (80 mmol/L sodium citrate, 52 mmol/L citric acid and 180 mmol/L glucose). After obtaining platelet-rich plasma, platelets were washed and incubated with 2.5 mmol/L fura-2/AM (Molecular Probes) for 45 min at room temperature. Then, after washing out fura-2, platelets were stored at room temperature in the dark until Ca^2+^ measurements were performed. Platelets were placed in fluorescence-free cuvettes (Sigma, Madrid, Spain) in the optical field of a fluorescence spectrometer (Aminco Bowman 2; Microbeam, Barcelona, Spain) and were excited alternatively with light at 340 and 380 nm, and the light emitted at 510 nm was collected. Changes in cytosolic free Ca^2+^ concentration ([Ca^2+^]i) were obtained using the fura-2 340/380 fluorescence ratio and calibrated as previously described [27]. Only one concentration of each drug was tested on every platelet suspension. The calibration procedure was done in every experiment to take into account differences in the number of platelets between animals. After obtaining baseline values for 30 s, the appropriate drug concentration was added and the fluorescence recorded. Three protocols were performed:The response to thrombin (0.1 and 0.3 U/mL) was studied in the absence (no Ca^2+^ added plus 0.5 mmol/L EGTA) and presence (1 mmol/L CaCl_2_) of extracellular Ca^2+^.Capacitative calcium entry (CCE) was determined by stimulating platelets in a Ca^2+^-free medium with thapsigargin (1 mmol/L). After 180 s, Ca^2+^ (1 mmol/L) was added, and the changes in [Ca^2+^]i were monitored for another 180 s.Ca^2+^ accumulation into intracellular stores was estimated by suspending platelets in a Ca^2+^-free medium (+100 µmol/L EGTA) and challenging with ionomycin (5 µmol/L) and thapsigargin (1 µmol/L).

Statistical analysis: The data are expressed as the mean ± SEM. In order to compare the responses between groups, the AUC (area under the curve) of the individual Ca^2+^ responses was calculated as the integral of the rise in [Ca^2+^ ]i for 180 s above basal, taking a sample every second, and expressed as arbitrary units. Then, these values were used to compose the graphs made out of histograms, reflecting the mean of six individual experiments per group. The resulting values, as well as baseline values, were compared by one-way analysis of variance, and a Student-Newman-Keuls post-hoc test was conducted. A probability level of *p* < 0.05 was considered to indicate a significant difference.

## Figures and Tables

**Figure 1 ijms-24-10948-f001:**
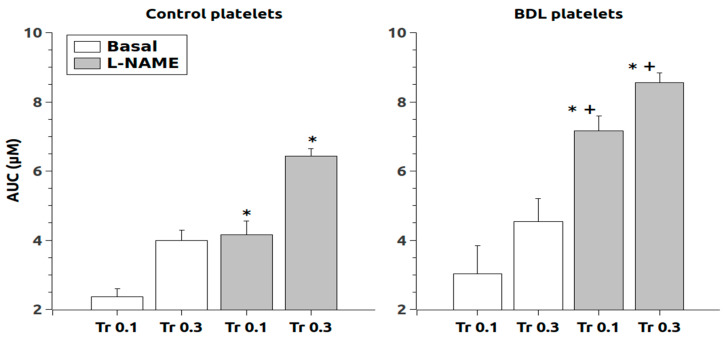
Area under the curve (AUC) of the Ca^2+^ responses after thrombin administration (0.1 and 0.3 U/mL) in the absence of extracellular Ca^2+^. Data were obtained in untreated platelets (basal) and after chronic treatment with L-NAME. Results are means + S.E.M. *, *p* < 0.05 vs. Basal; +, *p* < 0.05 vs. control.

**Figure 2 ijms-24-10948-f002:**
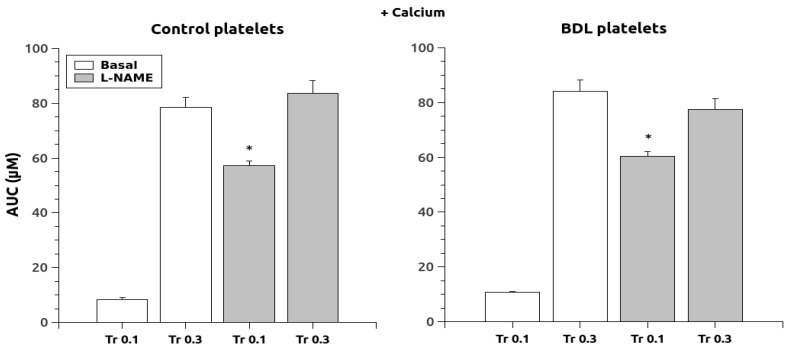
Area under the curve of the Ca^2+^ responses after thrombin administration (0.1 and 0.3 U/mL) in the presence of extracellular Ca^2+^. Data were obtained in untreated platelets (basal) and after chronic treatment with L-NAME. Results are means + S.E.M. *, *p* < 0.05 vs. Basal.

**Figure 3 ijms-24-10948-f003:**
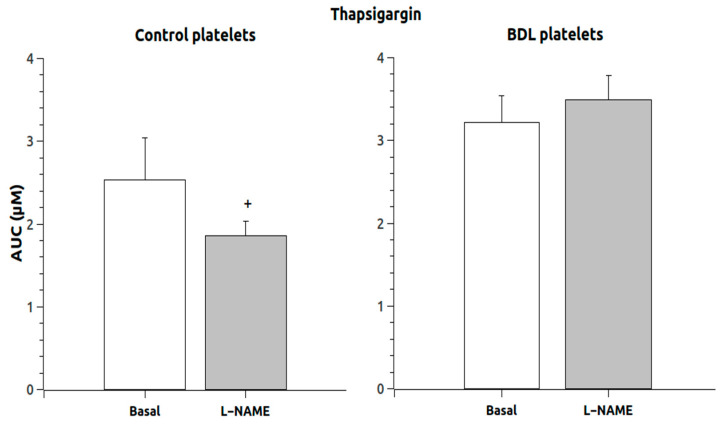
Area under the curve of the Ca^2+^ responses after thapsigargin administration in the absence of extracellular Ca^2+^. Data were obtained in untreated platelets (basal) and after chronic treatment with L-NAME. Results are means + S.E.M. +, *p* < 0.05 vs. control.

**Figure 4 ijms-24-10948-f004:**
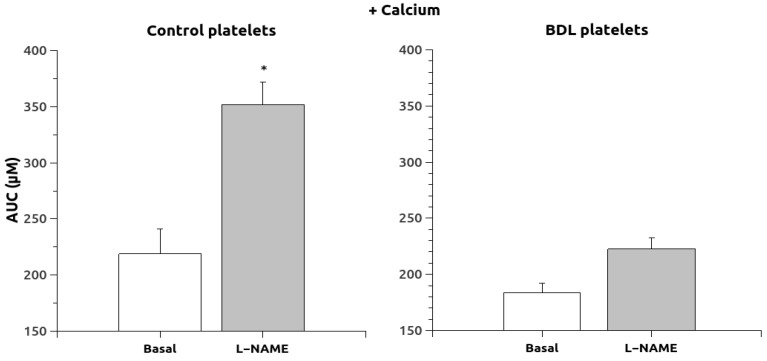
Area under the curve of the Ca^2+^ responses after calcium administration in thapsigargin-pretreated platelets, to allow for capacitative calcium entry. Data were obtained in untreated platelets (basal) and after chronic treatment with L-NAME. Results are means + S.E.M. *, *p* < 0.05 vs. Basal.

## Data Availability

Data are available from corresponding author on request.

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
