# Peer review of "Role of Nitric Oxide in the Altered Calcium Homeostasis of Platelets from Rats with Biliary Cirrhosis"

_ijms, 2023, doi:10.3390/ijms241310948_

Round 1

Reviewer 1 Report

In this manuscript, Akbari Aghdam and colleagues have performed a very well-designed, although simple, study in a rat model of liver disease.

I just would like to say that the Methods could be described in more detailed, specially those regarding the generation of graphs

Nevertheless, I would suggest transfer to another journal, since the amount of work performed is very low to be published in this journal.

English is understandable.

Author Response

We thank the reviewer for the comments.

We have expanded the methods section to include the obtention of blood from the animals an dalso included the number of animals, 6 per group (Subsection Isolation of platelets,...). 

We have also explained better the content in the Methods section (in subsection Statistical Analysis), specially the method used to calculate the area under the curve. 

Since these methods are well discussed in other manuscripts of our group, we did not want to fall into auto-plagiarism. However, we will be happy to expand more if the reviewer asks for it.

Reviewer 2 Report

The manuscript titled "Role of nitric oxide in the altered calcium homeostasis of platelets from rats with biliary cirrhosis" presents a study investigating the role of nitric oxide (NO) in the abnormal calcium signaling responses observed in platelets from bile-duct-ligated (BDL) rats. The authors conducted experiments on isolated platelets loaded with fura-2 and used fluorescence spectrometry to measure calcium levels. The results indicate that chronic treatment with the NO synthesis inhibitor L-NAME increased thrombin-induced calcium release from internal stores in both control and BDL rats, with a greater effect observed in BDL rats. Thrombin-induced calcium entry from the extracellular space was also elevated but at lower doses, and capacitative calcium entry was enhanced in control platelets but not in BDL platelets treated with L-NAME. The manuscript provides valuable insights into the role of nitric oxide in calcium signaling alterations in platelets from rats with biliary cirrhosis. However, there are some concerns that need to be addressed before publication.

Specific Comments:

  1. Abstract: The abstract provides a clear overview of the study, including the objective, methods, and key results. However, it lacks information on the sample size and statistical significance of the findings. Please consider including these details in the abstract.

  2. Introduction: The introduction provides a background on the platelet abnormalities in cirrhosis and the potential role of nitric oxide. However, it would benefit from a clearer and more focused research question or hypothesis that sets the stage for the study. Additionally, the introduction could be better organized and structured to guide the reader through the rationale and significance of the study.

  3. Methods: The methods section lacks crucial details about the experimental design, such as the number of animals or platelet samples used in each group. It is essential to include this information to ensure transparency and reproducibility of the study. Please provide details on the sample size and the number of replicates for each experimental condition.

  4. Results: The results section provides a clear presentation of the experimental findings, including figures and their corresponding descriptions. However, the presentation could be improved by providing the statistical significance of the observed differences and by indicating the specific statistical tests used.

  5. Discussion: The discussion adequately summarizes and interprets the study's findings in relation to previous research. However, it would benefit from a more comprehensive and critical analysis of the limitations and potential confounding factors of the study. Additionally, the discussion could explore the broader implications of the findings and suggest future research directions.

  6. Some sentences are lengthy and convoluted, making it difficult to follow the arguments. Additionally, there are grammatical errors and inconsistencies that need to be addressed.

  7. Resolution: Based on the issues raised, I recommend a resolution of "Reconsider after major revision (control missing in some experiments)." The manuscript has potential, but it requires significant revisions to address the mentioned concerns and improve the clarity and rigor of the research presented.

Overall, the study presents valuable insights into the role of nitric oxide in the altered calcium homeostasis of platelets from rats with biliary cirrhosis. Addressing the specific comments and making the necessary revisions will significantly strengthen the manuscript and increase its scientific rigor.

Author Response

We thank you the reviwer for the interesting comments.

  1. Abstract: we have included the number of animals used in each group, 6 per group. Regarding significance, we use the term "significantly" when there are statistically significant differences. Now, we have included the p level as well.
  2. The paragraph about the aim of the research has been completely rewritten to better express the objective of the work.
  3. The reviewer is right, sometimes we forget the most obvious things. We performed experiments in six animals per group, a total of 24. And in each one of these, all the experiments were performed, thus thrombin (0.1 and 0.3U) in the presence and in the absence of calcium, capacitative calcium entry and calcium accumulation in internal stores.
  4. We have added the p level to the text in the results section. Statistical test used (ANOVA) is mentioned at the end of the methods secttion. And, finally, the figure captions explains the symbols used to mark the significant differences with asterisks and crosses.
  5. We have added another paragraph at the end of the discussion to discuss that the use of the L-NAME inhibitor may be an influencing factor due to other effects in the nervous system and in the microcirculation.
  6. We are including a paragraph at the end of the discussion about the implications of the findings and suggesting also future investigations.

Round 2

Reviewer 1 Report

Although novelty is not very high, my comments have been addressed.

English is ok

Author Response

Thank you!

Reviewer 2 Report

The revised manuscript titled "Role of nitric oxide in the altered calcium homeostasis of platelets from rats with biliary cirrhosis" has addressed several concerns raised in the previous review.

However, there are still a few areas that require improvement:

1. Emphasize the potential limitations and confounding factors more explicitly. 

2. Discussion: Further highlight the broader implications of the findings and provide specific suggestions for future research directions. 

By addressing these specific suggestions for minor revisions, the manuscript will be further improved, ensuring clarity and accuracy in presenting the research findings.

Author Response

  1. Emphasize the potential limitations and confounding factors more explicitly. We have extended the paragraph to include this one: 

    This is a limitation of the present study and more studies with more specific inhibitors should be performed to better dissect the roles of the different NO isoforms. Other problems are related to the fact that the experiments were done in isolated and washed platelets which, although they are very frequently used in the laboratory setting, may interfere with the cell physiology. Thus, it would be interesting to perform other studies for instance, in whole blood, to study platelet function aggregation by means of light optical aggregometry, platelet-related primary hemostasis function using the PFA-100 System or the Multiplate Analyzer System (29).  Also, the use of intravital microscopy to study platelet aggregation in vivo may be another useful technique to apply in order to circunvent the problems related to in vitro techniques.

  2. Discussion: Further highlight the broader implications of the findings and provide specific suggestions for future research directions.  We have rewritten the last paragraph of the discussion and we hope that the referee likes it. The paragraph reads:  

    Thus, the findings of this study have important implications for understanding the mechanisms that underlie platelet dysfunction in liver cirrhosis and its potential impact on thrombosis and bleeding. The results indicate that NO plays a crucial role in defective calcium signaling observed in platelets of cirrhotic animals, which could contribute to the hypercoagulable state seen before ascites appears. However, these results do not have a direct translation to the clinics since they are of experimental nature, and thus, because of this, these conclusions should be taking with caution. Nevertheless, more studies would be necessary in order to go deep into the mechanism behind the capacitative calcium entry, for instance by analyzing the STIM1-Orai1 protein complexes that mediate the altered calcium response that our study has identified. Also, future investigations could aim to further elucidate the precise mechanisms by which NO affects calcium handling in platelets, as well as to explore the contribution of both eNOS and iNOS isoforms to these effects. Additionally, future studies should consider examining other factors beyond L-NAME inhibition that may be contributing to these responses, such as BDNF levels or vasopressin changes. Another important area of research would be the study of the release of platelet vesicles and its components, specifically the role of its products released, which at the moment it is not known in liver cirrhosis. Other studies in whole blood would be also interesting, as stated earlier, to eliminate the possible confounding interference of the experimental protocol with isolated and washed platelets. Finally, given the conflicting reports on platelet function in liver cirrhosis patients regarding their risk for both thrombosis and bleeding complications, future research using animal models can help clarify underlying pathophysiological mechanisms leading to altered hemostasis seen clinically.